# Win Big with Small: The Influence of Organic Food Packaging Size on Purchase Intention

**DOI:** 10.3390/foods11162494

**Published:** 2022-08-18

**Authors:** Shichang Liang, Ling Qin, Min Zhang, Yuxuan Chu, Lili Teng, Lingling He

**Affiliations:** 1School of Business, Guangxi University, Nanning 530004, China; 2Financial Research Center, Guangxi University, Nanning 530004, China; 3College of Economic and Management, Nanning Normal University, Nanning 530001, China

**Keywords:** organic food, packaging size, green perceived value, purchasing intention, construal level

## Abstract

People pay much attention to food and health issues, more so these days. Organic food brings its own “organic” aura as soon as it is produced. Despite the many studies on organic food packaging at present, they mainly focus on packaging design, materials, and colors and pay less attention to packaging size. In view of this gap in the literature, this study explores the influence of organic food packaging size on consumer purchase intention. This article conducted two experiments with 755 participants to examine the effect of organic food packaging size on purchase intention. The results show that the packaging size of organic food has a significant influence on consumer purchase intention. Specifically, the small size of organic food packaging (vs. large) can improve consumer purchase intention, and the green perceived value plays an intermediary role (Study 1). In addition, the consumers’ construal level moderates the influence of organic food packaging size on their purchase intention. For consumers with a high construal level, the small size of organic food packaging (vs. large) can improve their purchase intention. For consumers with a low construal level, large packaging size (vs. small) of organic food can improve their purchase intention (Study 2). This study reveals the psychological mechanism and boundary conditions of organic food packaging size on consumer purchase intention and provides practical enlightenment for enterprises in formulating the size of organic food packaging.

## 1. Introduction

Today, a growing number of people pay attention to food and health issues [1,2]. Organic food brings its own “organic” aura as soon as it is born. From processing to serving and then to eating, organic food is without pesticide, pollution, and fertilizer, thus earning the consumers’ notice. China’s organic food consumption market is growing at an annual rate of 25%, with a strong growth momentum. How to grasp the “hot spot” of environmentally friendly organic food and make organic food a frequent visitor at the table is a concern for the government and businesses. 

Product packaging is a “silent salesman”; it is also a common concern of merchants [3]. Manufacturers and retailers use product packaging as a visual communication medium and an important means to attract consumer attention [4,5]. Different from ordinary foods, organic foods are certified foods, and consumers pay more attention to the external clues to the products, so product packaging is particularly important for the purchase of organic foods [6,7]. An increasing number of merchants have noticed that product packaging size has an important influence on consumers’ decision making [8]. For example, when hedonic goods appear in the form of small package size, consumers are less affected by self-control and tend to consume more; when hedonic goods appear in large packaging sizes, consumers are more likely to think carefully and consume less [9]. Does packaging size also affect the consumers’ purchase behavior of organic food? This study answers this question.

The existing literature on organic food packaging can be summarized into two aspects, namely, linguistic cues (e.g., labels, nutrient composition tables, etc.) and non-linguistic cues (e.g., shapes, colors, patterns, materials, etc.) [10]. In terms of language cues, to improve human health and safety and reduce environmental hazards, the federal government regulates food labels [11]. Studies show that the presence or absence of organic food labels affects the consumers’ purchase behavior [12]. Compared with foods with only organic food labels, consumers give higher evaluation to foods with organic food labels and other detailed information [13]. Among the information included on the packaging of health food, consumers pay attention to nutritional value, food composition, and country of origin [14]. In terms of non-verbal clues, compared with that of traditional products, the packaging design of organic products often uses white and green images to show their naturalness [15]. To identify organic products on supermarket shelves more quickly, Paunonen et al. [16] suggested that standard dark green should be used consistently to indicate organic. Tu et al. [14] took middle-aged and elderly people as research objects and summarized the “happiness design” of organic food packaging. Among non-verbal cues, food packaging size, as an intuitive packaging element, affects the consumers’ emotional cognition and purchase intention [17]. However, despite the attention people pay to organic food, the influence of organic food packaging size on consumers’ purchasing behavior is less studied. Sultan et al. [18] proposed that organic food packaging size may be a factor affecting consumer decision making, but the relationship between organic food packaging size and purchasing behavior is not clearly explained. Therefore, this study explores the influence of organic food packaging size on consumers’ purchase intention and makes up for the gap in the existing literature.

Organic food is environmentally safe food, produced by environmentally sound methods, does not involve modern synthetic inputs such as pesticides and fertilizers, does not contain genetically modified organisms, and is not processed using irradiation, industrial solvents, or chemical food additives [19]. Organic food is generally considered healthier and more environmentally friendly than ordinary food [20]. Package size is widely used as an important marketing tool to promote consumption [21,22]. Studies show that products with small packaging size can improve the consumers’ quality perception [23]. Therefore, compared with the large packaging size, the small packaging size of organic food may better convey the signal of the high-quality characteristics of organic food to consumers. The high-quality signal (healthier, more environmentally friendly, and safer) can then enable consumers to obtain the green perceived value (GPV) of organic food and ultimately affect their purchase intention. In addition, according to construal level theory [24], consumers with a high construal level are more likely to pay attention to product interest demands, whereas consumers with a low construal level are more likely to pay attention to product attribute demands [25]. Specifically, consumers with a high construal level are more likely to pay attention to the interest demand that the small packaging size of organic food (compared with the large packaging size) has higher green value, which leads to higher purchasing intention. However, consumers with a low construal level pay more attention to the product attributes of large packaging size, large quantity, large discount, rather than the product benefits of small packaging size and high green value, so large packaging size (vs. small) can improve their purchase intention of organic food.

This study combines the packaging size of organic food with the construal levels of the consumer to explore the effect of food packaging size on consumer purchase intention. This study has three contributions. First, this study expands the research on packaging size in organic food. Despite the many studies on organic food packaging at present, they mainly focus on packaging design, materials, and colors and less on packaging size. Second, this study enriches the research on the intermediary role of GPV. The effect of organic food packaging size on purchase intention is clearly mediated by GPV. Finally, this study extends construal level theory to the field of organic food packaging. That is, the construal level affects consumer purchase intention for organic food with different packaging sizes.

## 2. Literature Review and Research Hypothesis

### 2.1. Product Packaging and Organic Food

The two uses of product packaging are product protection [26] and product circulation [27,28]. In this study, packaging pertains to all products made of materials of any nature in the process from raw materials to processed products and from producers to users or consumers, that are used for the closure, protection, handling, delivery, and display of goods [27]. Packaging, as an important marketing communication tool, affects the consumers’ perception of products and helps consumers make choices from many commodities [29,30,31]. Currently, there are three main research directions in the field of packaging marketing: the characteristics and functions of packaging [32], effect of packaging on consumers [33], and integration of packaging and other fields [34,35]. According to the clue attributes reflected by product packaging, it can be divided into two categories: language clue and non-language clue. The language clues of packaging include product information, manufacturer, country of origin, brand and so on [10]. Non-verbal clues of packaging include packaging graphics, colors, sizes, forms, materials, and so on [10]. At present, research on packaging size can be divided into two perspectives: consumers and enterprises. From the consumer perspective, scholars have discussed the relationship between packaging size and consumption [21], and unit cost is the key factor in this relationship [36]. Other topics discussed are the relationship between unhealthy food packaging size and food intake [37,38,39], consumer responses to package size reduction [40,41], and the sensitivity of consumers to changes in packaging size and price [42]. From an enterprise perspective, packaging size is an important tool for enterprises to participate in market competition. Packaging size plays a critical role in changing product unit price [43], product promotion [44], and product positioning [45].

According to the classification of packaging by Butkevičienė et al. [10], the research on organic food packaging can be summarized into two aspects: linguistic cues and non-linguistic cues. For language cues, the focus of discussion is the role of organic food packaging labels and their effect on consumer behavior [13]. In addition, consumers have higher purchase intention and lower calorie estimation for products with inorganic food labels [46]. However, consumers’ cognition of organic food labeling is based on subjective rather than objective knowledge [47]. For non-verbal cues, the sustainability of packaging materials has a positive effect on the perceived quality of food [48]. Healthy foods with sustainable packaging materials are considered fuller than the same foods packaged in non-sustainable packaging [49]. In terms of packaging color, Schuldt [50] pointed out that the green nutrition label on packaging increases people’s perception of health. In addition, Seo et al. [51] found that participants in Korea were more willing to buy properly packaged organic biscuits than over-packaged organic biscuits. However, in the research of packaging elements, few scholars discussed the packaging size of organic food and the purchase intention of consumers. Package size is a non-verbal clue in consumer choice and purchase, which affects consumer perception and emotional preferences [10], psychology and behavior [52]. Among the non-verbal cues, packaging size has a significant effect on the consumers’ purchase decisions compared with other elements [53]. Therefore, studying the influence of organic food packaging size on consumer behavior is important.

### 2.2. Packaging Size and Organic Food

Packaging is the most intuitive judgment basis for consumers. Despite much text information on food labels, consumers seldom use this information; instead, they use non-text information to evaluate food [54]. On the one hand, when the information is incomplete, people have to bypass the knowledge gap to make judgments [55]. On the other hand, non-verbal information obtained through external clues from the food packaging is easier to identify and remember than verbal information [56,57]. Thus, consumers are more inclined to use psychological shortcuts to draw conclusions about products according to the external characteristics of packaging (e.g., size, color, etc.) [55]. Consumers usually weigh before buying products and then make a decision whether to buy or not. Perceived value is an overall evaluation after consumers weigh the perceived benefits and costs of products or services [58]. Consumers’ perceived value is a subjective concept [59], which is often not the actual value of products, but the surplus or loss after comparing the actual value with consumers’ psychological perception. Perceived value is the overall evaluation of product quality by consumers [60]. When consumers buy organic food, the understanding of organic food-related knowledge makes consumers’ cognitive level relatively low [61], and consumers cannot use enough knowledge to judge the perceived value of buying organic food. At this time, consumers use a heuristic information processing mechanism, simplify the decision-making process by using packaging as an external clue, and make perceived value judgments quickly based on limited information [62]. Compared with products with large packaging sizes, consumers have a higher perceived quality for products with small packaging sizes [23,63] and, thus, have higher perceived value [64]. Perceived value is a significant predictor of consumer purchase intention [65]. The higher the perceived value, the stronger the consumers’ purchase intention [66]. Therefore, this study infers that the packaging size of organic food will affect consumer purchase intention.

Perceived value is not a one-dimensional concept but a multi-dimensional concept. Sheth et al. [67] divided it into five dimensions: social value, emotional value, functional value, cognitive value, and situational value. With the rise of environmental protection awareness, Chen and Chang [68] put forward the concept of GPV, and defined it as the overall evaluation of consumers’ net benefits between benefits and contributions of products or services according to environmental needs, sustainable development expectations, and green needs. GPV, as defined by Chen and Chang [68], is a one-dimensional concept, which is a new development of a perceived value dimension under green consumption. According to the multiple dimensions of perceived value, GPV is developed into a multi-dimensional concept that includes functional value, social value, emotional value, and situational value [69,70]. The GPV in this study is from the perspective of environmental protection and green consumption, so it is consistent with the single-dimensional GPV of Chen and Chang [68]. In the study of the relationship between perceived value and consumer purchase intention, Zeithaml et al. [58] pointed out that product quality does not directly affect consumer purchase behavior but affects the consumers’ purchase behavior through perceived value as an intermediary variable. Kuo et al. [71] also found that service quality has no direct and obvious influence on repurchase intention, but it indirectly affects repurchase intention through perceived value and customer satisfaction. Therefore, for organic food, using GPV as an intermediary variable to explain the effect of organic food packaging size on consumers’ purchase intention is appropriate.

At present, the problem of food waste is becoming increasingly serious [72], which is accompanied by a wide range of environmental impacts, such as soil erosion, deforestation, water and air pollution, and greenhouse gas emissions [73]. In the samples used by Williams et al. [74], 20–25% of food waste is related to packaging design attributes, and the damage caused by food waste to the environment has exceeded that of food packaging itself [75]. Adjusting packaging size can effectively reduce food waste, especially when the average household population continues to become smaller [76]. Organic food with small packaging size is more in line with actual demand. When food waste is reduced, the cost of social environmental governance is correspondingly reduced [77]. That is, small packages are more in line with the demands of environmental protection. Therefore, compared with large package size organic food, consumers will have higher GPV for small package size organic food. For organic food with large packaging size, the possibility of garbage disposal will increase because of incomplete consumption [78]. Compared with small package size, research shows that merchants adopting large package size is more beneficial to environmental protection because less packaging materials are needed to distribute each unit of food [79]. With the increasing use of sustainable packaging materials, packaging materials themselves are becoming more environmentally friendly [80], but the environmental problems caused by food waste are growing more prominent. Therefore, compared with large package size of organic food, the consumers’ GPV of small package size on organic food is higher.

Purchase intention is the sum of all motives behind a certain behavior of consumers [81]. Perceived value positively affects consumers’ purchasing attitude [82], which is a significant predictor of consumers’ purchasing intention [65]. Nowadays, environmental protection has become an important driving force for consumers’ green consumption behavior [83]. Janssen [47] used household panel data in his analysis. He found that natural health and environmental protection are the two most crucial driving factors for consumers to buy organic food. Laureti and Benedetti [84] pointed out that people who pay attention to environmental protection information, such as animal welfare, soil pollution, and deforestation, are more likely to buy organic products in their daily lives. Generally, environmental protection is an important factor for consumers to buy organic food; it is also the main embodiment of the perceived net benefits of green value [68]. Therefore, the higher the consumers’ GPV, the more consumers can meet the pursuit of environmental protection and sustainable development, and the stronger the consumers’ willingness to buy. Compared with large package size organic food, small package size gives consumers higher green perception value, and it can better meet the consumers’ pursuit of environmental protection and sustainable development of organic food. Moreover, consumption does not increase when the important purchase criteria do not match the cognitive beliefs about the purchase target [85]. In other words, compared with small package-sized organic food, a large package size is more likely to cause food waste and environmental pollution, so consumers like to buy organic food with environmental protection attributes. A mismatch exists between the important purchase criteria and the cognitive belief of the target product, so the consumer’s purchase intention will not increase. Hence, consumers will have a higher purchase intention for the small package size of organic food (vs. the large package size) with higher GPV.

To sum up, Hypotheses 1 and 2 are deduced as follows:

**Hypothesis 1** **(H1):**
*Compared with large packaging size, small packaging size of organic food can improve consumers’ purchase intention.*


**Hypothesis 2** **(H2):**
*GPV mediates the influence of organic food packaging size on consumers’ purchase intention.*


### 2.3. Packaging Size and Construal Level

Construal level theory is a “purely cognitive-oriented” social cognitive theory [86]. People’s reaction to social events depends on their psychological representation, which can be divided into high construal level and low construal level. Individuals with high construal level tend to process information in a general, core, abstract, and de-background way that reflects events [87]. Individuals with low construal level tend to process information in an accidental, peripheral, concrete, and contextual way [87]. For example, locking the door is understood by individuals with low construal level as “locking the door with a key,” whereas individuals with high construal level understand it as “anti-theft” [88]. Different psychological distances affect people’s construal level, which in turn affect their judgment and behavior [87].

Products based on interest demands are more attractive to consumers with a high construal level than attribute demands, whereas products based on attribute demands are more attractive to consumers with a low construal level than interest demands [25]. Product attributes are intrinsic characteristics attached to products, and their characteristics are measurable, concrete, observable, and related to differences from substitutes, such as physical characteristics and quantitative characteristics [89]. Product benefit refers to the conceptual exact value obtained by consumers through the consumption or possession of a product, such as convenience and happiness [89]. Specifically, when consumers have a close psychological distance and a low construal level, they are more likely to pay attention to characteristics such as product price [90]. When consumers have a distant psychological distance and a high construal level, they are more likely to pay attention to characteristics such as product quality value [86]. For consumers with a high construal level, organic food with small package size is more attractive because organic food with small package size is generally considered to be more conducive to reducing waste, reducing environmental pollution caused by food waste, and helping environmental protection and sustainable development, all of which satisfy consumers’ interest demands for environmental protection of products [91]. Compared with a large package size of organic food, small package size is more environmentally friendly, so it has higher GPV [68], and, thus, promotes consumers to have higher purchase intention. For consumers with a low construal level, organic food with large packaging size is more attractive because products with large packaging size are usually considered to be favorable for a large number of people, have the characteristics of low price and large product content, and satisfy consumers’ demands for product attributes [92]. Consumers with a low construal level have weak interest demands for a small package size of organic food, but they have strong attribute demands for a large package size of organic food. The demand for product attributes with large quantity is more attractive to consumers with a low construal level than the demand for product interests to protect the environment. Therefore, these consumers pay more attention to the product attributes of large packaging size and large quantity discount of organic food rather than the product benefits of small packaging size and high green value of organic food. For them, large packaging size (vs. small) can improve their purchase intention of organic food.

To sum up, product interest demands are more attractive to consumers with a high construal level. Compared with large packaging size, small packaging size of organic food can bring higher GPV, satisfy consumers’ interest demands, and then help to improve their willingness to buy organic food with small packaging size. Product attribute demands are more attractive to consumers with a low construal level. Compared with organic food with small packaging size, consumers with a low construal level have higher purchase intention for organic food with large packaging size.

To sum up, hypothesis three is put forward.

**Hypothesis 3** **(H3):**
*The construal level moderates the influence of organic food packaging size on consumers’ purchase intention.*


**Hypothesis 3a** **(H3a):**
*For consumers with a high construal level, compared with large packaging size, small packaging size of organic food can improve their purchase intention.*


**Hypothesis 3b** **(H3b):**
*For consumers with a low construal level, large packaging size of organic food can improve their purchase intention than small packaging size.*


## 3. Study 1: Influence of Organic Food Packaging Size on Purchase Intention

### 3.1. Purpose of the Study

In this study, Wuchang (a famous Chinese food brand) organic rice, a real brand from “Firewood Courtyard”, was used as a stimulus to verify H1 and H2.

### 3.2. Pre-Test

To test whether Wuchang organic rice from “Firewood Courtyard” selected in Study 1 is generally considered as an organic food and the effectiveness of packaging size (large vs. small) manipulation, a pre-test was conducted. A total of 184 participants were recruited to participate in the pre-test, of which 115 were female (62.5%), with an average age of 28.8 years.

The researchers first gave the participants a brief introduction about Wuchang organic rice from “Firewood Courtyard.” Wuchang organic rice from “Firewood Courtyard” originated from the “Golden Rice Farm” in Wuchang, Northeast China. The rice has organic certification, which adopts manual transplanting and weeding, and does not use pesticides and fertilizers. Drying ears naturally in the sun can prolong the ripening cycle of rice, and the taste is soft, waxy and mellow. Two groups of participants were then shown pictures of Wuchang organic rice in large package size (5 kg) and small package size (2.5 kg). In this study, a common express carton in life was selected as a reference, and rice with large and small packaging sizes was placed in the carton (see Appendix A) so as to manipulate the packaging size. Then, participants’ recognition of Wuchang organic rice as organic food and their perception of the packaging size of the product were measured. To exclude alternative explanations of other factors, this study also measured participants’ brand familiarity, product familiarity, and brand affection. The results showed that the two groups of participants recognized Wuchang organic rice as organic food (M large package = 6.31, SD = 0.748; M small package = 6.11, SD = 0.698; t (180) = 0.039, *p* > 0.05), and there was no significant difference. Significant differences in the perception of organic food package size were noted between the two groups (M large package = 4.86, SD = 1.251; M small package = 3.60, SD = 1.449; t (180) = −6.212, *p* < 0.05) as well as in brand familiarity (M large package = 5.25, SD = 1.607; M small package = 4.98, SD = 1.565; t (180) = −1.159, *p* > 0.05) and product familiarity (M large package = 5.27, SD = 1.547; M small package = 5.10, SD = 1.555; t (180) = −0.711, *p* > 0.05), and brand affection (M large package = 5.92, SD = 1.084; M small package = 5.68, SD = 1.018; t (180) = −1.530, *p* > 0.05). Moreover, the rationality of the experimental stimuli was guaranteed effectively. In conclusion, the manipulation of organic food and its packaging size (large vs. small) was successful.

### 3.3. Main Experiment

#### 3.3.1. Subject and Design

Study 1 used a single factor inter-group design; the independent variable was organic food packaging size, the dependent variable was purchase intention, and the mediator variable was GPV. A total of 296 participants were recruited in a university in Southern China, 140 were women, accounting for 47.3%, with an average age of 21.3 years. After the experiment, each participant was randomly paid 5–10 yuan. All participants were randomly assigned to large- and small-package-size groups.

#### 3.3.2. Experimental Process and Variable Measurement

The researchers told participants that the purpose of this study was to do product market research. All participants were randomly assigned to two groups: large package size and small package size. The researchers first gave the participants a brief introduction about Wuchang organic rice from “Firewood Courtyard”, then showed pictures of the product, and then answered relevant questions after the participants read the stimulating materials. Stimulation materials and manipulation tests were exactly the same as the pre-experiment. For the measurement of intermediary variables, refer to the GPV scale (Cronbach’s α = 0.933) [68], with a total of five measurement items (7 subscales, 1 = disagree; 7 = strongly agree): “The environmental function of Wuchang rice from ‘Firewood Courtyard’ provides me with very good value”; “The environmental protection performance of Wuchang rice from ‘Firewood Courtyard’ meets my expectations”; “I bought Wuchang rice from ‘Firewood Courtyard’ because it is more concerned about environmental issues than other products”; “I bought Wuchang rice from ‘Firewood Courtyard’ because it is very environmentally friendly”; “I bought Wuchang rice from ‘Firewood Courtyard’ because it provides more environmental benefits than other products.” The purchase intention of the organic food scale (Cronbach’s α = 0.908) developed by Grunert et al. [93] and Schifferstein and Ophuis [94] was used to measure the dependent variable. The scale included three items (7 subscales, 1 = disagree; 7 = strongly agree): “If I can buy ‘Firewood Courtyard’ Wuchang organic rice in the supermarket, I will buy it”; “I will buy ‘Firewood Courtyard’ Wuchang organic rice, although its price is higher”; “It is very possible for me to buy Wuchang organic rice in ‘Firewood Courtyard’.” At the same time, the control variables such as brand familiarity, product love, hunger during the survey, and whether there was a clear weight loss goal were measured. Demographic variables such as age and gender of participants were recorded.

### 3.4. Data Analysis

#### 3.4.1. Manipulation Test

The two groups of participants thought that Wuchang organic rice from “Firewood Courtyard” belonged to organic food (M large package = 6.00, SD = 1.195; M small package = 6.12, SD = 1.189; F (294) = 0.770, *p* > 0.05). The researchers also looked into their perception of package size (M large package = 4.81, SD = 1.401; M small package = 4.39, SD = 1.623; F (294) = 5.648, *p* < 0.05), brand familiarity (M large package = 3.73, SD = 2.121; M small package = 4.11, SD = 1.990; F (294) = 2.505, *p* > 0.05), product familiarity (M large package = 3.79, SD = 2.107; M small package = 4.07, SD = 1.845; F (294) = 1.520, *p* > 0.05), and product preference (M large package = 4.73, SD = 1.487; M small package = 4.91, SD = 1.277; F (294) = 1.281, *p* > 0.05). No significant difference was noted between the two groups regarding whether there was a clear weight loss goal (M large package = 4.33, SD = 1.943; M small package = 4.11, SD = 1.956; F (294) = 0.968, *p* > 0.05). The results showed that the organic food and its packaging size were successfully manipulated.

#### 3.4.2. Hypothesis Testing 

ANOVA for purchase intention. Taking packaging size as an independent variable and purchasing intention as a dependent variable, variance analysis was carried out. The results showed that packaging size of organic food significantly affected consumers’ purchasing intention. Compared with large packaging size, small packaging size of organic food could improve consumers’ purchasing intention of Wuchang organic rice from “Firewood Courtyard” (M large package = 4.45, SD = 1.536; M small package = 4.83, SD = 1.199; F (294) = 5.449, *p* < 0.05). The result supported H1 (see Figure 1).

#### 3.4.3. Mediating Effect of GPV

Compared with the large-package-size group, the small-package-size group had higher GPV (M large package = 5.18, SD = 1.304; M small package = 5.52, SD = 1.127; F (294) = 5.965, *p* < 0.05). In this study, with the size of organic food packaging as independent variable, GPV as intermediary variable, and consumer purchase intention as dependent variable, the bootstrap method was adopted to test intermediary effect, using PROCESS Model 4 of Hayes [95]. The iterative sampling times was set to 5000 times. The results showed that the indirective effect did not contain 0 (LLCI = −0. 4779, ULCI = −0. 0540) at 95% confidence interval, and the mediating effect was −0.261. The mediating effect of GPV was significant. The results supported H2.

### 3.5. Discussion

Study 1 verified the effect of organic food packaging size on consumers’ purchase intention. Compared with large packaging size, consumers have higher GPV for organic food with small packaging size and have higher purchase intention. No significant difference was noted in brand familiarity, product familiarity, product love, clear weight loss goal, and hunger degree between the two groups, which excluded the interference of alternative explanations of these variables. However, although H1 and H2 were verified by using Wuchang organic rice, a real brand organic food in “Firewood Courtyard,” the influence of participants’ inherent cognition of the real brand on the experimental results was not eliminated. In addition, rice usually appears in a large package size, which leaves the inherent impression of consumers and may affect the experimental results. In view of this, Study 2 selected cabbage of the virtual brand “Green Life” to further test H1 and H2 as well as the moderating effect of construal level.

## 4. Study 2: Effect of Construal Level

### 4.1. Purpose of the Study

To eliminate the influence of participants’ inherent cognition of a real brand, Study 2 took a cabbage of the virtual brand “Green Life” as a stimulus to further verify the H1 and H2. The moderating effect of construal level (H3) was also verified.

### 4.2. Pre-Test

Study 2 used the virtual organic food brand “Green Life.” The experimental stimulus was changed from Wuchang organic rice from “Firewood Courtyard” to cabbage from the virtual brand “Green Life” (see Appendix B). A total of 198 participants were recruited for pre-test, among which, 122 were women, accounting for 61.6%, with an average age of 28.8 years. The researcher told the participants that a market survey was being conducted in this study, and read the following to the participants: “‘Green Life’ Food Company was established in Nanning, Guangxi in 2019. The company is mainly engaged in organic agricultural products, such as vegetables, rice, and miscellaneous grains. Now, the products have been sold all over the country. This year, the company launched a new product ‘Green Life’ cabbage. We insist on using high-quality water and soil in the planting process. ‘Green Life’ cabbage is minimally processed to maintain its integrity, and the food contains no artificial ingredients, preservatives, or radiation. The food has reached the certification standard of organic food.” A transparent plastic box was selected as a reference, and “Green Life” cabbages with large packaging size (500 g) and small packaging size (250 g) were put into the transparent plastic box so as to control the packaging size. Subsequently, participants’ recognition of “Green Life” cabbage as organic food and their perception of the packaging size of the product were measured by a 7-level scale. To exclude alternative explanations of other factors, this study also measured participants’ brand familiarity, product familiarity, and brand affection. The results showed no significant difference on identifying with organic food between the two groups (M large package = 6.09, SD = 0.905; M small package = 6.29, SD = 0.746; t (196) = 1.715, *p* > 0.05). The two groups of participants perceived organic food package size, and the large package size scored higher than small package size (M large package = 3.96, SD = 1.558; M small package = 2.90, SD = 1.313; t (196) = −5.179, *p* < 0.05). No significant difference was noted in brand familiarity (M large package = 3.90, SD = 1.723; M small package = 4.16, SD = 1.633; t (196) = 1.101, *p* > 0.05), product familiarity (M large package = 4.05, SD = 1.820; M small package = 4.33, SD = 1.846; t (196) = 1.085, *p* > 0.05), and brand affection (M large package = 5.32, SD = 1.194; M small package = 5.39, SD = 1.194; t (196) = 0.417, *p* > 0.05) between the two groups, which effectively ensured the success of the experimental stimuli.

### 4.3. Main Experiment

#### 4.3.1. Subject and Design

A total of 153 participants were recruited from the online platform, 92 were women, accounting for 60.1%, with an average age of 28.9 years. Study 2 used 2 (organic food packaging size: large vs. small) × 2 (construal level: high vs. low) between group experiment. Differently from Study 1, to eliminate the inherent cognitive influence of participants on real brands, organic food with virtual brands were used as stimulants in Study 2.

#### 4.3.2. Experimental Process and Variable Measurement

The researchers told participants that the purpose of this experiment was to carry out product market research. Participants were randomly assigned to four groups: 2 (organic food packaging size: large vs. small) × 2 (construal level: high vs. low). Construal level manipulation drew on the method of Freitas, Gollwitzer, and Trope [96]. First, participants were asked to read a picture about exercise and then describe “why” or “how” to exercise in a short paragraph of more than 20 words so as to activate participants’ high construal level and low construal level, respectively. Focusing on why something should be done leads to a high-level, abstract mindset, whereas focusing repeatedly on how something should be done leads to a low-level, concrete mindset. Then, the subjects were asked to answer whether they pay more attention to “why” or “how” to exercise. In this way, the manipulation test of construal level was carried out. For the control of large and small package sizes, participants first read the introduction of “Green Life” brand and its organic cabbage products and were shown pictures of “Green Life” cabbage products with large and small package sizes. At the end of the experiment, participants were asked to answer the corresponding variable items, such as manipulation items and purchase intention items, which were the same as those in Study 1. Demographic variables such as age and gender of participants were recorded.

### 4.4. Data Analysis

#### 4.4.1. Manipulation Test

One-way ANOVA showed that participants in the high construal group thought they were more concerned about why than those in the low construal group (M high construal = 3.83, SD = 1.684; M low construal = 3.23, SD = 1.621; F (151) = 4.959, *p* < 0.05). Compared with the high construal group, the participants in the low construal group thought they were more concerned about how (M high construal = 2.93, SD = 1.668; M low construal = 3.62, SD = 1.487; F (151) = 7.281, *p* < 0.05). Thus, the manipulation of the construal level was successful. The two groups of participants both considered “Green Life” cabbage as organic food (M large package = 6.12, SD = 0.794; M small package = 6.22, SD = 0.858; F (151) = 0.639, *p* > 0.05). The perception of package size in both groups showed significant difference (M large package = 3.92, SD = 1.384; M small package = 2.92, SD = 1.334; F (151) = 20.739, *p* < 0.05). The results show that the organic food and its packaging size were successfully manipulated.

#### 4.4.2. Hypothesis Testing

Purchase intention: two-way ANOVA was conducted. The result showed that the main effect of packaging size was significant (F (149) = 4.062, *p* < 0.05), but the main effect of construal level was not significant (F (149) = 0.175, *p* > 0.05). The interaction between packaging size and construal level was significant (F (149) = 25.142, *p* < 0.05). A simple effect analysis of the interaction showed that under the condition of high construal level, the purchase intention of small organic food package size was significantly higher than that of large organic food package size (M large package = 5.32, SD = 0.644; M small package = 6.09, SD = 0.628, F (74) = 28.003, *p* < 0.05); Under the condition of low construal level, the purchase intention of large organic food package size was significantly higher than that of small organic food package size (M large package = 5.82, SD = 0.662; M small package = 5.49, SD = 0.774, F (75) = 4.034, *p* < 0.05)(see Figure 2). The results verified H3.

Analysis of mediated effect: To further test the moderating effect of construal level and the mediating effect of GPV, this study used the bootstrap method (Model 7) of Hayes [96] to test the moderating mediated effect. The iterative sampling times were set to 5000 times. The packaging size of organic food was regarded as an independent variable (small packaging size was coded as 0, large packaging size was coded as 1), GPV was regarded as an intermediary variable, purchase intention was regarded as a dependent variable, and construal level was regarded as an adjustment variable (high construal level was coded as 0, low construal level was coded as 1). Under 95% confidence interval, the mediating effect of construal level on the influence of packaging size on purchase intention of organic food did not include 0 (LLCI = 0. 2751, ULCI = 0. 9712), and the effect size was 0.594, which showed that the mediating effect of construal level was significant. Specifically, for consumers with a high construal level, the mediating effect of GPV on the influence of organic food packaging size on purchase intention did not include 0 (LLCI = −0.6853, ULCI = −0.1799), and the effect size was −0.4117, which showed that the mediating effect was significant. This result also indicated that the mediating effect was significant.

### 4.5. Discussion

To eliminate the inherent cognitive influence of participants on real brands, Study 2 used cabbage of virtual brand “Green Life” as stimulus to avoid the interference of participants’ familiarity and love of existing brands and further verify the main effect. Specifically, organic food with small packaging size (vs. large) will enhance consumers’ GPV and significantly enhance their willingness to buy. At the same time, Study 2 verified the boundary condition of this effect, that is, construal level moderates the effect of organic food packaging size on purchase intention. Specifically, compared with large packaging size, small packaging size of organic food can improve the purchase intention of consumers with a high construal level (H3a). Larger packaging size of organic food can improve the purchase intention of consumers with a low construal level (H3b) than smaller packaging size.

## 5. General Discussion

### 5.1. Conclusions

At present, despite there being many studies on organic food packaging, they mainly focus on the label information, design, material, and color of product packaging and pay less attention to the size. This research aims to address this gap by studying the effect of organic food packaging size on purchase intention. Across two studies, using both field and experimental data, this research provides robust evidence that organic food packaging size can influence consumer purchase intention. Study 1 showed that compared with large packaging size, small packaging size of organic food can improve consumer purchase intention, and analyzed the intermediary role of GPV. Study 1 also excluded alternative explanations of product familiarity, brand familiarity, and brand liking. These effects were robust whether organic food packaging size was manipulated with the bag of organic rice (Study 1) or the box of organic cabbage (Study 2), and whether the brands were real or virtual. Study 2 showed the moderating effect of construal level, that is, compared with large packaging size, small packaging size of organic food can improve the purchase intention of consumers with a high construal level. Large packaging size of organic food can improve the purchase intention of consumers with a low construal level than small packaging size.

### 5.2. Theoretical Contribution

First, the research on packaging size was extended to the field of organic food. The previous research on organic food packaging can be divided into two aspects: linguistic cues and non-linguistic cues. The studies mainly focused on product packaging label information, design, materials, and colors [13,48] and paid less attention to packaging size. However, packaging size, as an important external feature of packaging, can affect consumer judgment and behavior choice of products. Therefore, this study on the effect of organic food packaging size on consumer purchasing behavior makes up for the lack of research regarding the non-linguistic cues of packaging and their influence on organic food-purchasing behavior.

Second, this work expands the research on the intermediary mechanism of GPV in organic food-purchasing behavior. The discovery of this intermediary mechanism not only uncovers the psychological mechanism of the consumers’ intention to buy organic food but also supplements the research on the psychological mechanism of the consumer intention to buy organic food. This study provides a theoretical basis for effectively carrying out organic food marketing activities and correctly guiding the consumers’ intention to buy organic food.

Third, this study enriches the research on organic food packaging by incorporating construal level. People with high construal level pay attention to the interest demands of products, whereas those with low construal level pay attention to the attribute demands of products [25]. Kim et al. [97] applied this finding to the selection of product promotion strategies. However, this study applies this conclusion to the organic food packaging field and explores consumer purchase intention on packaging sizes of organic food under different construal levels. Doing so not only addressed the gap that organic food packaging size is a neglected factor but also explores the boundary of construal level on the effect of organic food packaging size on the consumers’ willingness to buy.

### 5.3. Management Implication

First, these findings provide useful insights for food manufacturers to optimize the selection of organic food packaging size. As an important marketing tool, packaging has long been used by managers to boost sales. The optimal choice of packaging size can not only reduce costs, but also increase sales and gain profits. For organic food, our findings suggest that merchants can use the tool of organic food packaging size and adopt small packaging sizes for organic food with a lower unit price so as to enhance consumers’ GPV of organic food and offset the “low value sense” with low pricing. For organic food with a higher unit price, large packaging can be used more so that consumers can feel the “large quantity and benefits” of organic food and reduce the psychological barriers caused by the high price of an organic food purchase.

Second, our results provide insights for marketers to develop marketing strategies and tactics. On the one hand, it is possible to educate consumers about the benefit of organic food and to increase consumers’ green perceived value of organic food in general, thereby promoting an overall increase in organic food sales. On the other hand, compared to the large packaging size of organic food, the small packaging size of organic food gives consumers higher green perceived value, which provides consumers with more environmental value. However, large packaging sizes of organic food are considered more favorable than small packaging sizes of organic food. Therefore, marketing managers can set up a green boutique area and a special discount area in an organic food supermarket. In the green boutique area, organic food is positioned higher and focus on delivering higher green perceived value, that is, environmental value; while in the special discount area, it provides a special price to consumers and plays a role in promoting product sales.

Third, this research also highlights boundary conditions at the construal level that may help marketers to further improve the effectiveness of their marketing strategies. People with high construal levels prefer a small package size of organic food and pay attention to interest demands. However, people with low construal levels prefer organic food with a large package size and pay attention to attribute demands. Therefore, consumers can be induced to have a low or high construal level by changing the marketing environment during the marketing promotion. For small package size organic food, more publicity is carried out on the benefits appeal of organic food, to improve consumers’ pursuit of green perceived value. At the same time, when designing guiding language, words such as “you” and “I,” which show closer psychological distance, are used less, and words such as “he” and “she,” which show farther psychological distance, are used instead to increase consumers’ psychological distance and guide consumers’ high construal level. Doing so improves consumers’ GPV of organic food. In addition, organic food with the large packaging size can be promoted, and publicity can be carried out from the attribute appeal to enhance the promotion effect.

Fourth, organic food packaging size is one of the organic food packaging elements. When enterprises carry out marketing activities, they can also use it together with other packaging elements, analyze the characteristics of organic food concretely, and choose the most suitable organic food packaging design and marketing schemes.

### 5.4. Limitations and Future Research

There are three limitations in this study. First, organic food packaging size uses rectangular packaging shape. Whether organic food is packaged using non-rectangular shapes, such as round or trapezoidal, is not further discussed. Second, the construal level does not consider the influence of cultural factors. Self-awareness can not only be regarded as a state in a specific situation but also as a stable personality trait. Oriental people emphasize the relationship between people, and their psychological distance is close; Westerners emphasize independence, and their psychological distance is distant [98]. Future research can bring cultural factors into the research framework of organic food and further explore the differences and connections between the state construal level caused by situational factors and the trait construal level formed by cultural factors as well as their influence on consumption behavior. Third, many psychological factors and situational factors that can affect a consumers’ willingness to buy organic food, such as state subsidies and advocacy [99], health goals [100], and so on. However, this research only selects the GPV and construal level to discuss the intermediary and moderating effects. Moreover, a number of factors may be interacting with one another to affect the purchase intention of consumers for organic food. Fourth, there is a lack of a comprehensive model to guide the research. Future research should be built around an overarching theory, such as the theory of planned behavior, which provides a theoretical framework for analyzing the relationship between personal beliefs and behavioral predictions. The theory is also commonly used in research on consumer behavior and can nicely provide a holistic view of consumer behaviour [101,102]. Therefore, combining a more comprehensive model for theoretical construction, as such a theory of planned behaviour can be used as a future research direction.

## Figures and Tables

**Figure 1 foods-11-02494-f001:**
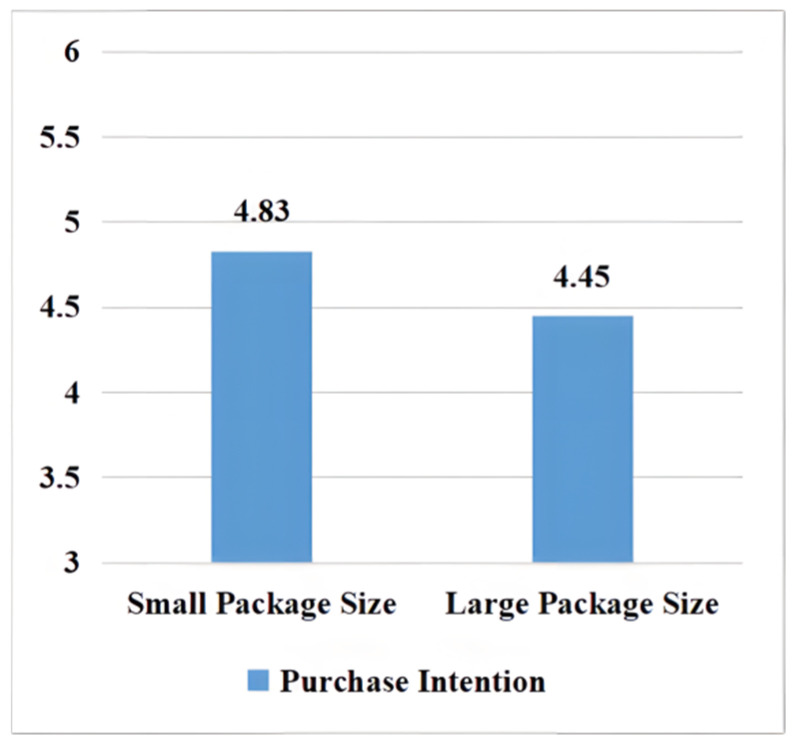
The influence of organic food packaging size on purchase intention.

**Figure 2 foods-11-02494-f002:**
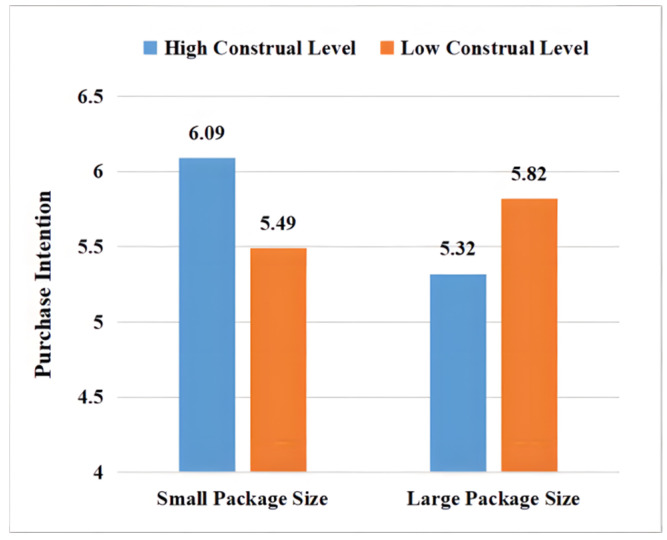
Influence of packaging size and construal level of organic food on purchase intention.

## Data Availability

The data presented in this study are available on request from the corresponding author. The data is not publicly available due to the need to maintain the confidentiality of study participants.

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
