# Peer review of "Win Big with Small: The Influence of Organic Food Packaging Size on Purchase Intention"

_foods, 2022, doi:10.3390/foods11162494_

Round 1
Reviewer 1 Report
The abstract contains main required components and it forms coherent text with logical conclusions and interactions between its immanent parts. The title of the paper is well formulated, and it covers the content. The introduction logically follows the aim of the paper and it provides valuable introspection into solved topic. Methodological part of the paper is suits current scientific standards. The appropriate statistical apparatus is described adequately. The results are presented clearly. The interpretation of tables and figures is acceptable. However, wider context of the presentation of the results should be applied to make the results really understandable for the audience. The level of the author’s knowledge is satisfying. However, I have few relevant comments on this paper:
- Authors would explain why they frame their analysis using the construal level theory, pointing out advantages and disadvantages of such theory compared to the many explaining consumer's behavior.
- Tables'readability could be improved, using colors and higher definitions.
- Marketing implication needs to be deeply expanded.
Author Response
Response to Reviewer 1 Comments
Point 1: Authors would explain why they frame their analysis using the construal level theory, pointing out advantages and disadvantages of such theory compared to the many explaining consumer's behavior.
Response 1: Thanks for your valuable comments. Construal level theory is chosen for analysis mainly because of the following advantages: First, construal level theory is well constructed and can explain a wide range of social and psychological phenomena. The construal level theory has now developed psychological distance into a unified theory that includes temporal distance, spatial distance, social distance, and hypotheticality. Construal level theory has been applied to many studies of consumer behavior. Second, construal level theory has different operational definitions, which is very suitable for the two studies of this article. In this study, referring to the existing literature, the construal level is operationally defined as the demand for the attributes of the product or the demand for the interest of the product(Hernandez and Wright et al., 2015). Third, the construal level can better explain the psychological mechanism related to the size of product packaging. When the psychological distance of consumers is far, consumers will have a high construal level and are more likely to pay attention to the attribute demands of products, such as reducing environmental pollution; and when the psychological distance of consumers is close, consumers will have a low construal level and are more likely to pay attention to the interest of products, such as lower prices. A package in small size is more environmentally friendly and has higher green perceived value than a package in large size, which can satisfy consumers' attribute demands(Williams and Wikström et al., 2012; Principato and Secondi et al., 2015); while a package in large size can satisfy consumers' interest demands for the special price. Therefore, consumers with a high construal level have a higher purchase intention for organic food with small package size; while consumers with a low construal level have a higher purchase intention for organic food with large package size. It has flexibility in theoretical application and enhances the rigor and explanatory power of theoretical derivation. However, this theory also has its disadvantages. Compared with other theories that explain consumer behavior, people will have different construal levels for differences in psychological distance. Therefore, the exploration of the reasons behind this is only people's empirical and intuitive explanations(Trope and Liberman, 2003).
Point 2: Tables'readability could be improved, using colors and higher definitions.
Response 2: Thanks for your valuable comments. We have made changes to the paper, adding color to the table, and adjusting the definitions to 300DPI.
Point 3: Marketing implication needs to be deeply expanded.
Response 3: Thanks for your valuable comments. According to your suggestion, we have deepened the marketing implication. First, we add a point of marketing implication. Our findings provide specific recommendations for marketers to develop marketing strategies and tactics. On the one hand, it is possible to educate consumers about the benefit of organic food and to increase consumers' green perceived value of organic food in general, thereby promoting an overall increase in organic food sales. On the other hand, compared to the large packaging size of organic food, the small packaging size of organic food gives consumers higher green perceived value, which provides consumers with more environmental value. However, large packaging sizes of organic food are considered more favorable than small packaging sizes of organic food. Therefore, marketing managers can set up a green boutique area and a special discount area in an organic food supermarket. In the green boutique area, organic food is positioned higher and focus on delivering higher green perceived value, that is environmental value; while in the special discount area, it provides a special price to consumers and plays a role in promoting product sales. Second, we adjust the other point of marketing implications. Please see the specific content in the part of the marketing implication.
The mentioned literature:
[1] Hernandez, J.M.D.C.; S.A. Wright; F. Ferminiano Rodrigues. Attributes versus benefits: The role of construal levels and appeal type on the persuasiveness of marketing messages. Journal of Advertising 2015, 44, 243-253, doi: 10.1080/00913367.2014.967425.
[2] Williams, H.; Wikström, F.; Otterbring, T.; Löfgren, M.; Gustafsson, A. Reasons for household food waste with special attention to packaging. Journal of cleaner production 2012, 24, 141-148, doi: 10.1016/j.jclepro.2011.11.044.
[3] Principato, L.; L. Secondi; C.A. Pratesi. Reducing food waste: an investigation on the behaviour of Italian youths. British Food Journal 2015, 117, 731-748, doi: 10.1108/BFJ-10-2013-0314.
[4] Trope, Y.; N. Liberman. Temporal construal. Psychological review 2003, 110, 403, doi: 10.1037/0033-295X.110.3.403.

Reviewer 2 Report
Is Wuchang rice from ‘Firewood Courtyard’ a well known brand? If yes, wouldn't this induce bias?
Section 3.3
There are sub-subheaders here? I'm unsure how the journal format this but this needs to be separated from the main paragraph - e.g., Subject & design
Study 1 - how did the authors control that it is indeed the small pack size, but not their consumption/purchase habit of buying smaller packs?
Study 1 & Study 2 utilises different statistical approach - why? If there are two factors in Study 2, wouldn't it be better to use 2way ANOVA to also understand the interactions?
An additional mediation analysis was performed, why? Is this really necessarily? The authors have stated several hypotheses in the introduction - why not follow that and see to prove the hypothesis is correct or not?
What exactly was the key highlight of this research? A more thorough discussion is needed here, meanwhile the literature review can be condensed further.
Author Response
Response to Reviewer 2 Comments
Point 1: Is Wuchang rice from ‘Firewood Courtyard’ a well known brand? If yes, wouldn't this induce bias?
Response 1: Thanks for your valuable comments. Wuchang rice from ‘Firewood Courtyard’ is a well-known regional brand in Heilongjiang Province in northeastern China. However, study 1 was conducted at a university in southern China, and most of the subjects were from the south. These subjects were less familiar with the regional brands in northern China, which could effectively reduce the experimental bias caused by brand awareness. At the same time, during the experiment, we measured brand familiarity and product familiarity. The results showed that the subjects were at a low level of brand familiarity (M = 3.92) and product familiarity (M = 3.93). And there was no significant difference between the two groups (p > 0.05).
Point 2: Section 3.3
There are sub-subheaders here? I'm unsure how the journal format this but this needs to be separated from the main paragraph - e.g., Subject & design.
Response 2: Thanks for your valuable comments. Such as Subject & design, etc. are used as sub-subheaders. We have separated them from the main paragraph as you suggested. Specific adjustments are shown in red.
Point 3: Study 1 - how did the authors control that it is indeed the small pack size, but not their consumption/purchase habit of buying smaller packs?
Response 3: Thanks for your valuable comments. Your comment is very informative. During the selection process of the experimental subjects, we selected the subjects randomly at a university in southern China, and the experimental groups were also randomly assigned. It has eliminated the influence of consumption habits and made subjects concentrate more on food package size through a variety of means. In addition, we did not discuss the habit of buying larger or smaller packages in the experiment mainly because these questions may make the subjects guess the purpose of the experiment and interfere with the experimental results. Finally, we have also noticed this when conducting the experiments, but the purchase habits themselves are also more complicated and are affected by factors such as family structure, income, and gender. Therefore, consumers' purchase habits and preferences for food packaging can be used as a new research direction. In the future, it is hoped that further research can be done to further expand the boundary conditions of this research.
Point 4: Study 1 & Study 2 utilises different statistical approach - why? If there are two factors in Study 2, wouldn't it be better to use 2way ANOVA to also understand the interactions?
Response 4: Thanks for your valuable comments. It may be the reason for our unclear statement, we used ANOVA for data processing in both study 1 and study 2, and used two-way ANOVA in study 2 to test the moderating effect. In order to make our data processing method more clearly understood by readers, we have re-adjusted this part in the manuscript, and the specific adjustments are shown in red.
Point 5: An additional mediation analysis was performed, why? Is this really necessarily? The authors have stated several hypotheses in the introduction - why not follow that and see to prove the hypothesis is correct or not?
Response 5: Thanks for your valuable comments. Study 1 verified H1 and H2, that is, the main effect and the mediating effect of green perceived value. The study 2 is to change the real brand into a virtual brand as an experimental stimulus, which verifies the H3, that is, the boundary condition of the construal level, and further tests the mediating effect of green perceived value. Therefore, our experiments are based on the logic of the hypothesis to carry out the experimental demonstration. At the same time, study 1 and study 2 verify the mediation twice, considering different experimental scenarios, to increase the external validity of the experiment.
Point 6: What exactly was the key highlight of this research? A more thorough discussion is needed here, meanwhile the literature review can be condensed further.
Response 6:
- Thanks for your valuable comments. With the increasingly prominent food and health issues, the consumption of organic food has attracted more and more attention. Organic food packaging is an important tool to promote organic food consumption. However, the existing literature on organic food packaging mainly focuses on food labels, colors, and materials, and less attention is paid to the size of organic food packaging(Hoogland and de Boer et al., 2007; Schuldt, 2013; Carmela and Ada et al., 2021). Food packaging size is one of the important packaging elements. It is of great significance for manufacturers and retailers to optimize packaging decisions, formulate correct marketing strategies and tactics, and improve market competitiveness by studying the impact of organic food packaging size on consumers' purchase Our research found that compared with the large size of organic food packaging, the small size of organic food packaging will make consumers have a higher purchase intention. Among them, green perceived value plays a mediating role (Study 1). In addition, we also explored the boundary conditions of the main effects, that is, the moderating role of the construal level (Study 2). For consumers with a high construal level, compared with the large packaging size, the small packaging size of organic food can improve their purchase intention; while for consumers with a low construal level, the large packaging size of organic food can improve their purchase intention than the small packaging size.
- We have simplified the three parts of the literature review in section 2.1. and deleted some of the detailed elaborations of the existing research results. Please see the specific content in section 2.1.
The mentioned literature:
[1] Hoogland, C.T.; J. de Boer; J.J. Boersema. Food and sustainability: do consumers recognize, understand and value on-package information on production standards? Appetite 2007, 49, 47-57, doi: 10.1016/j.appet.2006.11.009.
[2] Carmela, D.; M.B. Ada; R. Simona. The Effect of Sustainable Package on Taste Perception of Healthy Foods. Micro & Macro Marketing 2021, 2, 333-356, doi: 10.1431/99828:y:2021:i:2:p:333-356.
[3] Schuldt, J.P. Does green mean healthy? Nutrition label color affects perceptions of healthfulness. Health communication 2013, 28, 814-821, doi: 10.1080/10410236.2012.725270

Round 2
Reviewer 1 Report
I am fine with the current manuscript version.
Author Response
Response to Reviewer 1 Comments
Point 1: I am fine with the current manuscript version.
Response 1: Your insightful comments have been extremely helpful as we work to improve the paper. These issues will be given more attention in our future studies as well. We would like to express our sincere gratitude for your efforts. We also hope to receive more guidance from you in the future.

Reviewer 2 Report
The authors have addressed my comments. It is however important to note as a limitation of this study that future studies should include a more comprehensive model, as such a theory of planned behaviour (see ref: https://doi.org/10.3390/su13137430) should be utilised to provide a holistic view of consumer behaviour.
Author Response
Response to Reviewer 2 Comments
Point 1: It is however important to note as a limitation of this study that future studies should include a more comprehensive model, as such a theory of planned behaviour (see ref: https://doi.org/10.3390/su13137430) should be utilised to provide a holistic view of consumer behaviour.
Response 1: Thanks for your valuable comments. According to your suggestions, we have added some content to the section on future research. There is a lack of a comprehensive model to guide the research. Future research should be built around an overarching theory, such as the theory of planned behavior, which provides a theoretical framework for analyzing the relationship between personal beliefs and behavioral predictions. The theory is commonly used in research on consumer behavior and can nicely provide a holistic view of consumer behaviour [1, 2]. Therefore, combining a more comprehensive model for theoretical construction, as such a theory of planned behaviour can be used as a future research direction. Specific adjustments are shown in red. We will also take note of your recommendations and give this problem more consideration in our future studies and writing. Your insightful comments have been very helpful to us. We would like to express our gratitude for all of your efforts. We also look forward to receiving more advice from you in the future.
The mentioned literature
[1] Malavalli, M. M.; Hamid, N.; Kantono, K.; Liu, Y.; Seyfoddin, A. Consumers’ Perception of In-Vitro Meat in New Zealand Using the Theory of Planned Behaviour Model. Sustainability 2021, 13, 7430, doi: 10.3390/su13137430.
[2] Bîlbîie, A.; Druică, E.; Dumitrescu, R.; Aducovschi, D.; Sakizlian, R.; Sakizlian, M. Determinants of Fast-Food Consumption in Romania: An Application of the Theory of Planned Behavior. Foods 2021, 10, 1877, doi: 10.3390/foods10081877.
